# A Fast and Cost-Effective (FACE) Instrument Setting to Construct Focus-Extended Images

Gilbert Audira [1,†], Ting-Wei Hsu [2,†], Kelvin H.-C. Chen [3], Jong-Chin Huang [3], Ming-Der Lin [4,*], Tzong-Rong Ger [5] and Chung-Der Hsiao [1,6,7,*]

1   Department of Bioscience Technology, Chung Yuan Christian University, Taoyuan 320314, Taiwan
2   National Synchrotron Radiation Research Center, Hsinchu Science Park, Hsinchu 30076, Taiwan
3   Department of Applied Chemistry, National Pingtung University, Pingtung 900391, Taiwan
4   Department of Molecular Biology and Human Genetics, College of Medicine, Tzu Chi University, 701 Zhongyang Rd, Sec. 3, Hualien 97004, Taiwan
5   Department of Biomedical Engineering, Chung Yuan Christian University, Taoyuan 320314, Taiwan
6   Department of Chemistry, Chung Yuan Christian University, Taoyuan 320314, Taiwan
7   Research Center for Aquatic Toxicology and Pharmacology, Chung Yuan Christian University, Taoyuan 320314, Taiwan
*   Correspondence: mingder@gms.tcu.edu.tw (M.-D.L.); cdhsiao@cycu.edu.tw (C.-D.H.)
†   These authors contributed equally to this research.

**Abstract:** Image stacking is a crucial method for micro or macro photography. It captures images at different focal planes and then merges them into a single, all-in-focus image with extended focus. This method has been extensively used for digital documentation by scientists working at museums or research institutions. However, the traditional image stacking method relies on expensive instruments to conduct precise image stacking using a computer-based stepper motor controller. In this study, we reported how to conduct image focus extensions with comparable quality to those done by a motorized stepper using a cost-effective instrument setting and an efficient manual stacking method. This method provides a shorter operation time and capability to capture images of living objects and high flexibility in obtaining the images of objects from cm to mm scale. However, it also has some limitations, including the inability to control aperture and exposure time, relatively short working distance at high magnification, requires additional steps to convert the video into images, and heavily relies on the user's manual observation prior to a video recording. Nevertheless, the authors believe that the current method can be applied as an alternative method to conduct image stacking. The development of such an instrument and method offers a promising avenue for scientists to perform image stacking with greater flexibility and speed in macro photography.

**Keywords:** image stack; focus extension; diffusion tunnel





## 1. Introduction

Scientific photography is used for various practical purposes in zoology, botany, paleontology, and other fields of science. In addition, it is also commonly used in the conservation of art, history, and natural science specimens. Generally, these applications require the photographic documentation of objects in small sizes, ranging from centimeters to millimeters [1]. However, even though it can reveal unique patterns, textures, colors, and details unseen by the naked eye, macro photography still has some difficulties since it requires rigorous technique, perseverance, and patience [2]. Since the invention of photographic equipment, people have tried to digitize their collections. Thus, developing and curating high-resolution digital images are demanded as a standard for digital collections [3–5]. However, it becomes challenging to maintain the quality and get desired results in macro photography. Macro photography is commonly used for imaging subjects at reproduction ratios ranging from 1 to 5× magnifications [6]. Recent innovations such as

interchangeable lens camera systems and dedicated macro lenses make microphotography a cheaper alternative to capture images with desired magnifications [7–9].

The main barriers to a quality macro-photograph are losing the depth of field (DOF) and resolution. The low DOF makes it challenging to get the whole subject in focus [10]. Although the step-down approach in optical aperture could increase the DOF, aperture reduction could result in image aberration, distortion, and decreased optical resolution due to the diffraction effect [2,11,12]. Therefore, the photo-stacking technique is a better solution to extend DOF by montaging images captured at different focal planes into a single composite image with extended focal depth [12,13]. Currently, with advancing computational techniques, numerous software has been developed to perform photo-stacking and prevent image aberrations. These advancements enable photographers to maximize resolution using the highest possible aperture. Thus, digital macro photography can now achieve a single ultra-high resolution image generated from tens to thousands of overlapping images captured with the camera system's delicate, precise, and orthogonal movement [6,14]. However, photographic conditions must be optimized to obtain a high-quality stacked image. These include a high-resolution and high-quality macro lens that provides a high-resolution image with less chromatic aberrations or distortion, a motorized stepper that precisely captures images at different focal planes, and an external flash with a diffusion tunnel that provides high-intensity and uniform illumination. To advance digital macro photography, Longson et al. reported a giga-pixel method for image montage by using X, Y, Z, and F motorized stepper [15]. Moreover, Ströbel et al. developed an automatic motorized stepper for the 3D reconstruction of an insect [16]. *scAnt*, an open-source platform that consists of a scanner and a Graphical User interface, was also presented by Plum et al. to create digital 3D models of arthropods and small objects by enabling the automated generation of Extended Depth of Field images from multiple perspectives [17]. In addition, a new Fixed-Lens multi-focus image capture and a calibrated image registration technique using analytic homography transformation were also recently demonstrated to effectively apply an image-based 3D reconstruction of small-scale objects, including insects and biological specimens [18]. Unfortunately, these delicate setups are complex, expensive, and could be unaffordable to many research laboratories. As a result, several groups have designed simpler setups for image stacking to reduce the complexity and cost of macro photography. For example, Mertens et al. have reported a low-cost camera for entomological digitalization projects [19]. Brecko et al. have also demonstrated a semi-automatic instrument setting and compared the relative performance of several different software on stacked image quality and performance [20].

With the extensive usage of photographic illustrations in scientific fields, including scientific publication requirements, a novel method is always needed to help researchers document the subject faster and cost-effectively. However, to our knowledge, no research publication demonstrates a video-based method to efficiently acquire a stacked image, although Helicon Focus experimented with this method in 2015. This paper reports a rapid and cost-effective instrument setting for conducting video-based macro photography with an extended focus by assembling a fast and cost-effective (FACE) instrument that can conduct macro photography to extend focus depth by image stacking in the vertical position (Figure 1A). In addition, the results obtained by using the WeMacro instrument, a commercially available automatic focus-stacking rail, were also presented to provide a comprehensive side-by-side comparison with the current method (Figure 1B).

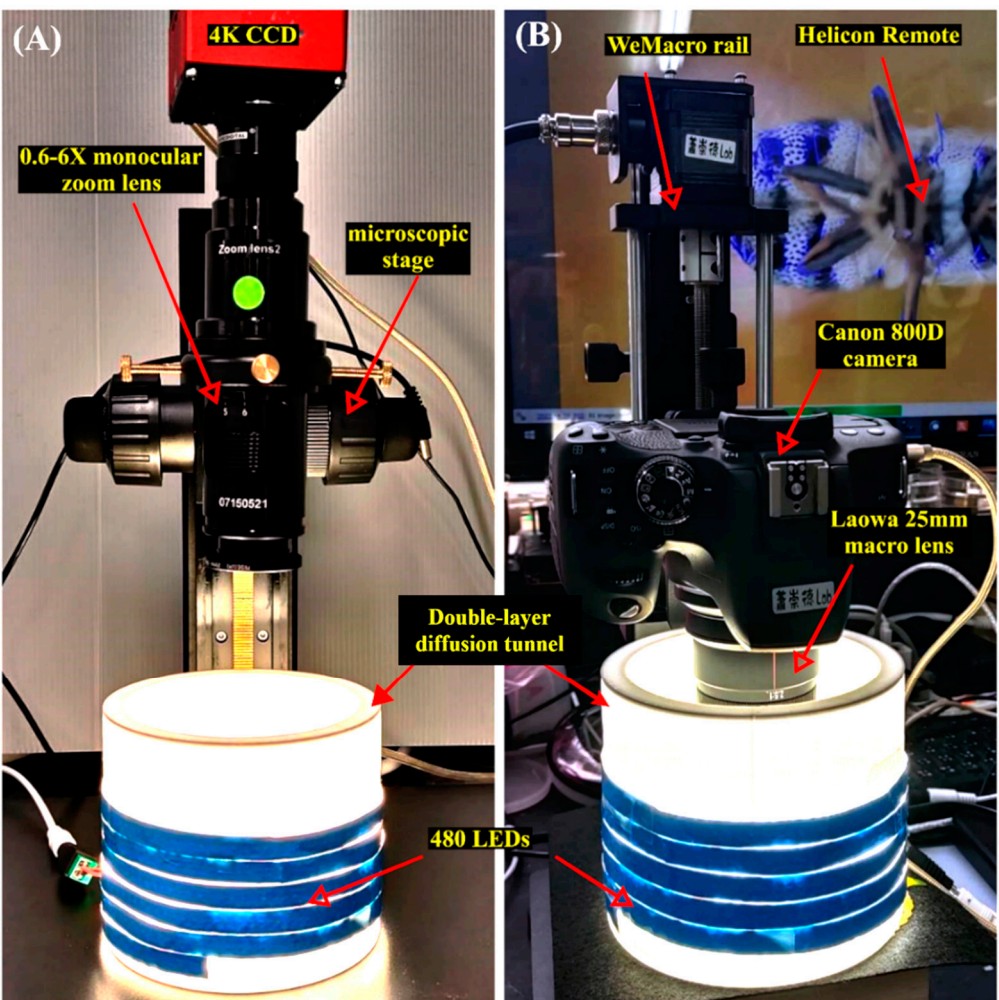

**Figure 1.** Overview of FACE and commercial motorized instrument settings used to conduct macro-photography in the vertical position in this study. (**A**) The instrument setup for FACE includes a 4K CCD, a microscopic stage, and a 0.6–6× monocular macroscopic lens. (**B**) The motorized stepper system is mounted with a Canon 800D digital camera equipped with a Laowa 25 mm 2.5–5× macro lens. For both methods, a 3D-printed double-layer diffusion tunnel was used to obtain uniform lighting.

## 2. Materials and Methods

### 2.1. The Setup of the FACE Instrument

A 4K CCD (HSJS Photoelectric instrument, Shenzhen, China) with a 1/1.7″ Sony sensor was mounted onto a zoom lens (ZGMzoom, Zgenebio, Taipei, Taiwan) with 0.6 to 6× magnification (Figure 1A). In addition, by using thread adapters, some macro lenses like DCR-150, DCR-250, MSN-202, and MSN-505 (Raynox, Tokyo, Japan) or microscopic objective lenses such as HC Plan 4× and 10× (Phenix Optics, Jiangxi, China) can also be mounted onto a 0.6 to 6× monocular microscopic zoom lens to provide additional magnification (Figure S1). This image/video recording device was later mounted onto a microscopic holder (HSJS Photoelectric instrument, Shenzhen, China) that enabled us to adjust its Z position. A homemade double-layer diffusion tunnel that contained 480 light-emitting diode (LED) bulbs was used as the light source to provide uniform illumination around the objects (Figure S2). For the first step of the current FACE system operation, the subject was fixed in the central position of a homemade double-layered diffusion tunnel. Later, the XY position of the subject was manually adjusted. After XY position alignment, the video recording was started while the macro photographic instrument was slowly moved in the Z-axis from the proximal to distal focus planes.

### 2.2. The Setup of the Motorized Stepper Instrument

The desktop computer was connected to a Canon 800D camera with an APS-C size sensor installed with a 3-piece macro extension tube set (31 + 21 + 13 mm, total 65 mm extension) and Laowa 25 mm ultra macro lens (Venus Optics, Hefei, China) with 2.5–5× magnification. Later, the camera was mounted on the WeMacro rail (https://www.wemacro.com/ (accessed on 29 March 2022), Shanghai, China) to enable the adjustment of the camera in the Z position by controlling the motorized stepper that was also connected to the desktop computer (Figure 1B). Afterward, Helicon Remote (version 3.9.12 W, https://www.heliconsoft.com/heliconsoft-products/helicon-remote/ (accessed on 29 March 2022)) was used to capture the stacked images of the objects automatically. In the current experiment, each object's applied total steps differed depending on the object's width with 20 μm of focusing steps. We also applied a 1-s pause after the movement of the stepper to minimize the vibration during the image capture process. In addition, we also used the lowest value of ISO (100) to reduce the image's noise. Finally, the captured images were saved on the computer desktop to be processed later. As the current FACE instrument method, a homemade double-layer diffusion tunnel illuminated by 480 LED bulbs was used to provide uniform illumination around the objects.

### 2.3. Construction of the Diffusion Tunnel

3D printing was conducted to produce a double-layer diffusion tunnel with a cylinder shape (Figure S2). The design file in .stl file type for the diffusion tunnel can be found in the Supplementary Materials. Later, a 2-m LED light strip with 480 LED bulbs (organized by a 2835 SMD LED chip) was twined into the cylinder and supplied with 12 V 5 A electrical power to maintain stable lighting conditions. In addition, to prevent potential melting problems caused by the intense heat generated during the LED illumination, the diffusion tunnel was 3D-printed with acrylonitrile butadiene styrene (ABS).

### 2.4. Orientation of Subjects and Background Replacement

Insect needles at 0 or 00 sizes were inserted into the insect body for better orientation adjustment. For the lateral view, the shaper end of the insect needle was inserted into the insects at the thorax segment, and the other end was embedded with clay. Rotating the insect needle angles can finely adjust the subject's position. For the frontal view, an insect needle was inserted into the subject from the distal to the proximal position (Figure S3). Materials such as natural leaves and non-woven fabric with colors with a size of 1.5 cm × 1.5 cm could be placed in between subjects and clay to provide a more uniform and colorful background (Figure S4).

### 2.5. Image Stacking

After the images were obtained, Helicon Focus (version 8.0.4, https://www.heliconsoft.com/heliconsoft-products/helicon-focus/ (accessed on 29 March 2022), Kharkiv, Ukraine) was used to do the image stacking process. First, videos and images captured by FACE and WeMacro rail instruments were loaded into Helicon Focus. For video files, Helicon Focus can automatically convert each video frame into an individual image. Later, we stacked those images into a single image with a wide focus by using this software. Helicon Focus provides three different methods of image stacking. Each image stacking method was tested for the object used in the present study; Pyramid Method (Method C) with smoothing value 1 gives the best-stacked image compared to other methods.

### 2.6. Computer Setup

A computer desktop with Intel Core i9-9900 K CPU @ 3.60 GHz processor, 32 GB RAM, and NVIDIA GeForce GTX 950 graphic card was used to run all of the necessary programs in the present study, including Helicon Remote and Helicon Focus.

### 2.7. Post-Editing of Images

After image stacking, the stacked image in JPG format was transferred to PhotoScape X (http://x.photoscape.org/ (accessed on 29 March 2022)) to enhance image sharpness, high dynamic range adjustment, color balance, or background replacement.

## 3. Results

### 3.1. Comparisons of Magnification Level between the Current FACE and Motorized Stepper Methods

Prior to the image quality comparison, it was intriguing to learn the magnification performance between the current FACE and motorized stepper methods. The original monocular zoom lens of the FACE method has 0.6–6×, while the Laowa 25 mm macro lens for the motorized stepper method has 2.5–5× magnification power. For better macro viewing, several macro or objective lenses were mounted onto a 0.6–6× zoom lens for the FACE system. Meanwhile, for the current motorized stepper system, three pieces of extension tubes with a total of 65 mm in length were mounted between the camera and Laowa macro lens to also obtain a higher magnification power. By measuring the field of view (FOV), the combinational usage of macro or objective lens in the FACE system offered a higher magnification level than the current motorized stepper method. For example, when the monocular zoom lens was set at 3× magnification, the usage of additional lenses significantly reduced the FOV from 6.8 mm to either 3.6 mm (DCR-150 + 250), 2.3 mm (MSN-202), 2.1 mm (4× objective lens), 1.8 mm (MSN-505), or 1.4 mm (10× objective lens). Meanwhile, when the current motorized stepper system was set at 3× magnification, the addition of a 65 mm extension tube reduced FOV from 7.6 mm to 4.1 mm with a gradual decline at higher magnifications. An extension tube in the current motorized stepper setting enhanced the magnification levels; however, it was less efficient than a macro lens mounted with an objective lens, as demonstrated in the current FACE setting (Figure 2). In conclusion, with the help of objective lenses, the current FACE system had a broader range of magnification levels compared to the current motorized stepper setup.

### 3.2. Comparisons of Working Distance between the Current FACE and Motorized Stepper Methods

Next, the working distance between the lens and subjects for both methods was also measured to investigate how much space was left between the front lens and the subject. For the current FACE system, the working distance from 2× to 6× magnification was maintained at 87 to 103 mm. However, after being mounted with a macro lens, the working distance sharply declined to either 40 mm (DCR-150 and DCR-250), 20 mm (DCR-150 + DCR-250 and MSN-202), or 15 mm (MSN-505). Furthermore, when microscopic objective lenses such as 4× or 10× magnification power were installed, the working distance sharply declined to 10 mm. On the contrary, the current motorized stepper system coupled with Laowa 25 mm macro lens maintained a consistent working distance of around 40 mm at all magnification levels (Figure 3). Therefore, although the current FACE system could reach very high magnification with smaller FOV, the relatively short working distance caused the operation to be less convenient compared to the current motorized stepper system. As a result, the operation of the FACE method at a high magnification power should be performed with caution to avoid a collision between samples and the lens.

### 3.3. Comparisons of Operation Time between the Current FACE and Motorized Stepper Methods

To evaluate the current FACE system performance, the operation time of this method was also comprehensively compared to the current motorized stepper method. To minimize variation, the same instrument setups, such as lighting, image stacking, and post-editing processes, were used in both methods, and thus, the differences between these two setups were only located in the image recording device and processes. For the current FACE method, the videos were captured by a monocular zoom lens mounted onto a 4K CCD, while for the motorized steeper method, a Laowa 25 mm macro lens was installed onto a Canon 800D camera and mounted in WeMacro rail, which can perform fully motorized

image capture controlled by Helicon Remote software. For the FACE method, manual operation conducted video recording of all of the current objects from proximal to distal focus planes for the subject within 5–10 s. However, the required time for the video recording process depends on the thickness of an object or the length of desired depth of field. If the video recording is too short, the number of obtained images will be too view to be stacked and might result in an out-of-focus image. On the other hand, although stacking more images can reduce the noise-to-signal ratio and provide more images to be stacked, stacking too many images will produce unnecessary images and increase the recording duration. Therefore, as one of the limitations of this method, exercise is required by the users to obtain a sense of how long video recording is needed for the desired focus-stacked scene. In obtaining similar images shown in this report, ~5 s of video recording is recommended for a 1-cm thick object, which in this case is an insect. Later, the video was uploaded onto Helicon Focus to extract 150–300 frames from the videos since the video output for this setting was at 30 fps. On the contrary, the current motorized stepper took around 3–5 min to finish the image capture, which highly depended on the width of the objects. In addition, ones have to keep in mind that, unlike the typical macro photography method that uses flash as an external light source, the present study used a diffusion tunnel to provide continuous and uniform lighting since by using this modified setup, a video recording with uniform lighting conditions was able to be conducted continuously.

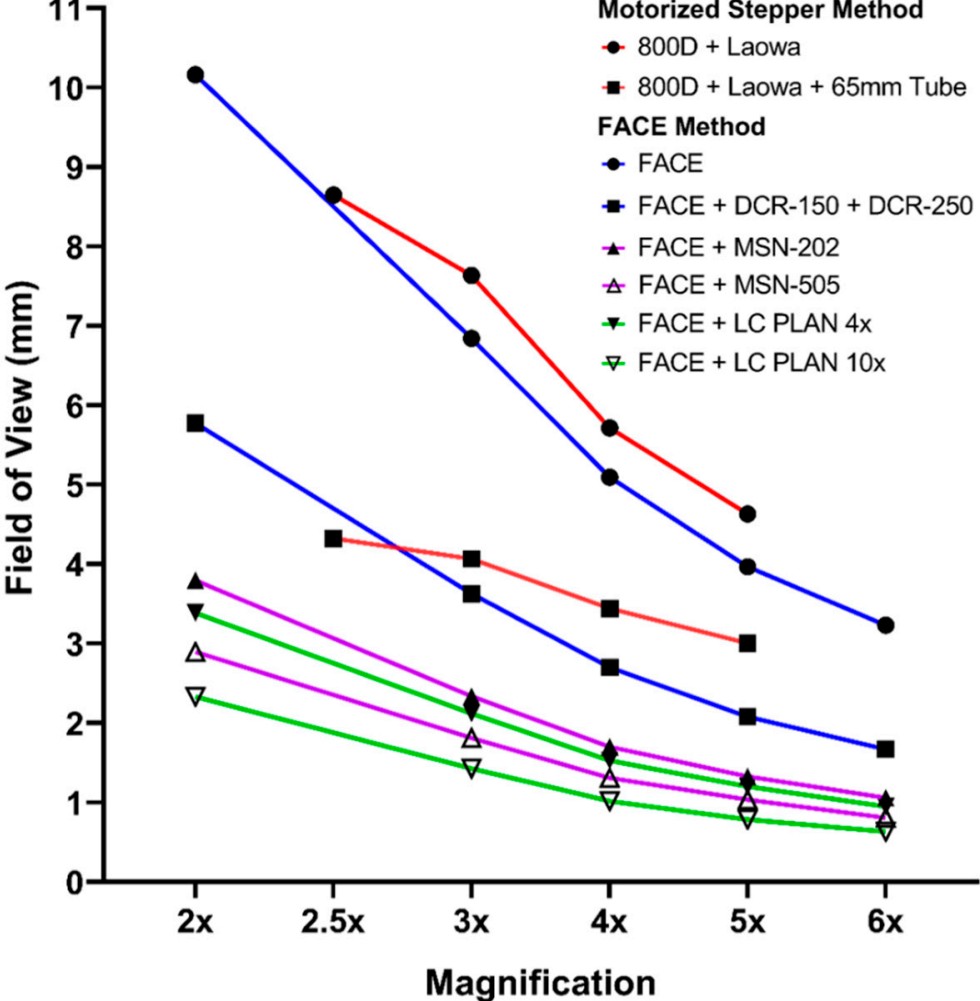

**Figure 2.** Comparison of the field of view (FOV) between the current FACE and motorized stepper setups with various lens combinations. The *X*-axis indicates magnification levels for the different photography systems. The *Y*-axis indicates the corresponding field of view (Abbreviation: 800D, Canon EOS 800D camera; Laowa, Laowa 25 mm macro lens. Tube, 65 mm extension tubes).

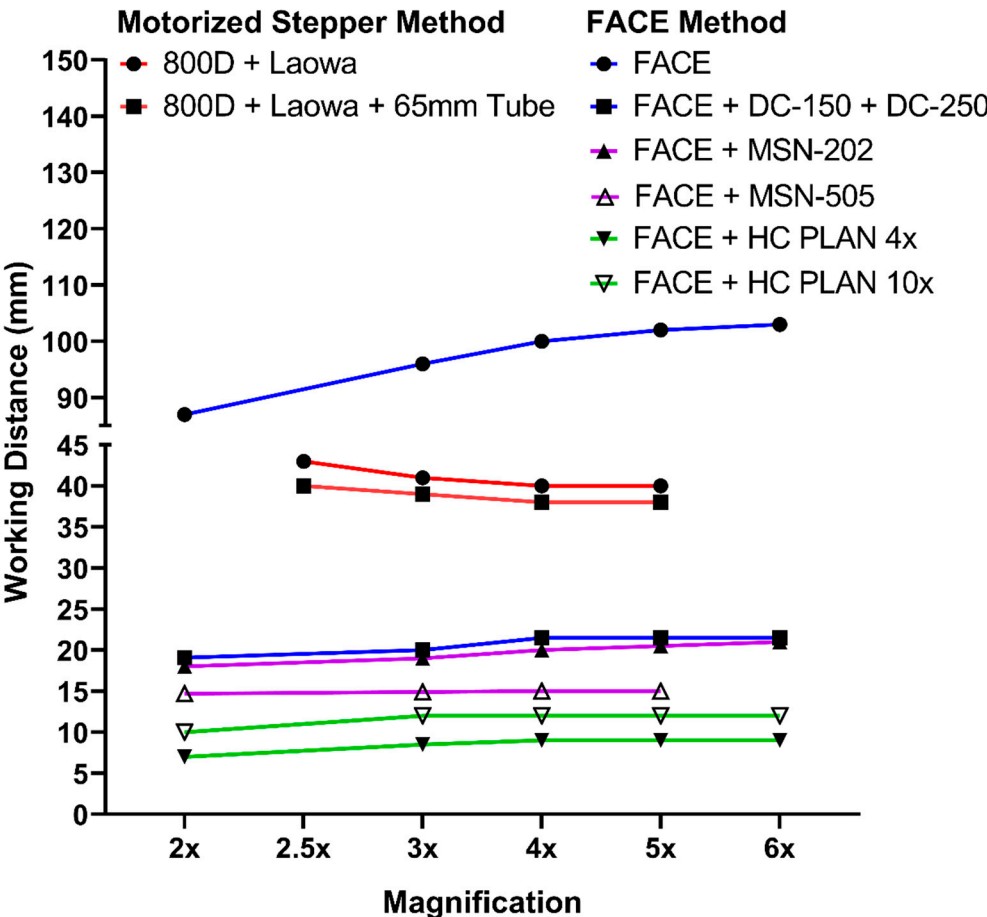

**Figure 3.** Comparison of the working distance between the current FACE and commercial motorized stepper setups with various lens combinations. The *X*-axis indicates magnification levels for different photography systems. The *Y*-axis indicates the corresponding working distance between the lens and objects. Abbreviation: 800D, Canon EOS 800D camera. Laowa, Laowa 25 mm macro lens. Tube, 65 mm extension tubes.

Finally, the image stacking quality from the current FACE method was evaluated by conducting a side-by-side comparison between this method and the commercial motorized stepper method. Here, two images with a comparable FOV from each method were chosen (Figure 4). For a fly with a body length of around 5 mm, a stacked image by the current FACE method with 2.5× magnification (FOV = 8.5 mm) could yield outstanding details in visualizing the entire body structure that was comparable to the current motorized stepper method that was set at 2.5× magnification (FOV = 8.6 mm) (Figure 4A,B). For visualizing the fly head, the FACE system was also set at 2.5× magnification, and a 4× objective lens (FOV = 2.8 mm) was mounted in it to achieve a stacking image with fine details (Figure 4C). Interestingly, the output image is comparable to that obtained from the motorized stepper method, which was also set at 2.5× magnification and connected with 65 mm extension tubes (FOV = 3.0 mm) (Figure 4D). Next, the performance of the current FACE method was also evaluated in a pharaoh ant (*Monomorium pharaonic*), a relatively smaller object with a body size of around 2 mm. By using the same setting as in capturing the fly head images, the current FACE method also yielded fine details (Figure 4E, FOV = 2.8 mm) that are also comparable to that obtained from the current motorized stepper method, which was set at 5× magnification and installed with 65 mm extension tubes (Figure 4F, FOV = 3.0 mm). Besides the ventral view, it is also intriguing to assess the image quality of the current FACE method in lateral and frontal views of several insects with complex face organs. For the lateral view, a hoverfly with a body size of around 4 mm was chosen as an object.

As we expected, a stacked image by the FACE method with 2× magnification that was enhanced with a 4× objective lens resulted in fine details image of its head (Figure 4G, FOV = 3.3 mm), which is similar to the image generated by the current motorized stepper with a 65 mm extension tube that was set at 4× magnification (Figure 4H, FOV = 3.4 mm). Lastly, a honeybee with a body size of around 15 mm was used for the image quality assessment from the frontal point of view. Similar to other results, the current FACE method without additional lenses, acquired a stacked image with fine details in the head part at 2.5× magnification. Meanwhile, an output image with an identical quality was generated by the current motorized stepper method that was set at 2.5× magnification (Figure 4J, FOV = 8.6 mm). In addition, one must remember that the image resolutions of these two methods were different, although the output images are comparable. While the current motorized stepper method outputs images with a resolution of 6000 × 4000 pixels (Canon 800D), the videos obtained from the current FACE method only have 3840 × 2160 pixels (4K CCD). Therefore, with the rapidity of technological advancement in the present day, there always will be room for improvements in the current FACE method. The results demonstrate that when sufficient and uniform lighting is provided, the stacked image's quality obtained from the current FACE system is comparable to those obtained from the motorized stepper system.

### 3.4. The FACE Method Is Cost-Effective

The costs of each method to conduct macro photography were listed and compared. The detailed cost for each component of each method is summarized in Table 1. Before moving further, one has to consider that the listed prices may vary depending on the region and the availability of the products. As shown in the table, the total cost of the current FACE method, including all essential instruments and macro lens to conduct macro photography, was estimated at around 800–1450 USD, whereas the motorized method cost about 1750 USD. For regular magnification from 0.6–6×, we recommend using a regular RACE setup, and the total cost will be 800 USD. For magnification higher than 6×, we recommend using additional microscopic or close-up lenses mounted onto the RACE setup, and the total cost will be around 1450 USD. After deeper investigation, the camera and the lenses are the most expensive components of both systems. However, even though the current FACE system employs a 4K CCD to record the videos at a frame rate of 30 fps, other cameras with a similar capability or even lower grade can also be used as an alternative if it is impossible to afford a recording device with a similar quality with the current CCD. Nevertheless, one must keep in mind that the stacked image quality may be downgraded if the recording devices cannot record 4K videos at 30 fps. Moreover, the current FACE system is also compatible with various lenses, such as zoom, objective, and macro lenses, which greatly expand its operational flexibility for macro photography by having a wide range of magnifications. Meanwhile, for the motorized stepper system, Canon 800D camera with an APS-C sensor was used to capture high-resolution images. To the best of our knowledge, the camera for this system is also interchangeable with other Canon cameras or even some Sony cameras, which may be expanded to other cameras, depending on the development of the Helicon Remote software in the future. To sum up, in terms of total cost in affording the system, the current FACE system outcompetes the current motorized stepper units and software.

## FACE method

## Motorized Stepper Method

**Figure 4.** *Cont.*

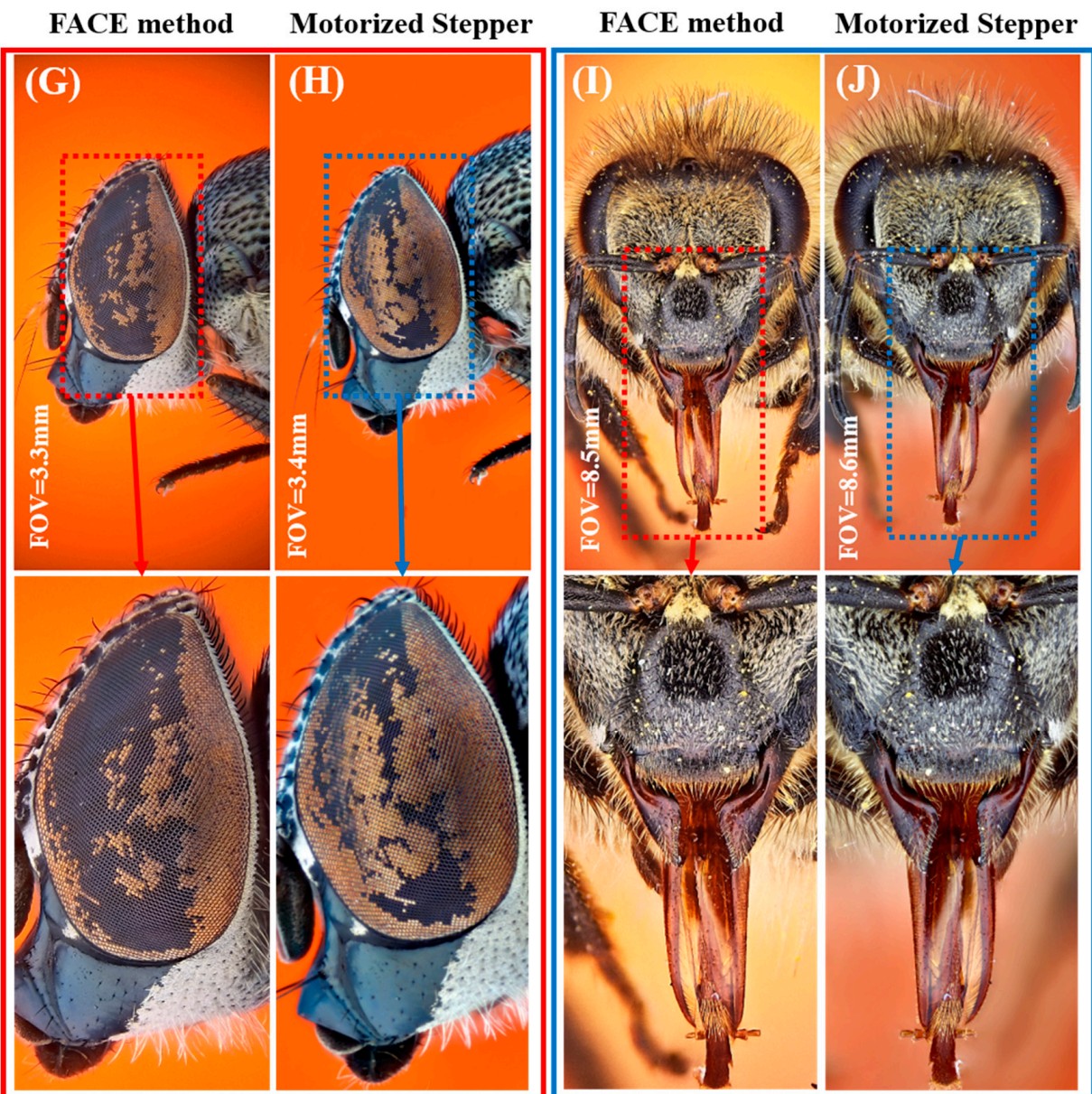

**Figure 4.** The comparisons of several stacked images of various insects shot at different orientations by FACE (red color) and motorized stepper (blue color) methods. (**A**,**B**) Ventral view images of fly stacked from 163 and 166 images captured by the FACE system and motorized stepper, respectively, with 2.5× magnification. (**C**,**D**) Ventral view images of the fly head stacked from 180 photos captured by the FACE system with a 4× objective lens at a 2.5× magnification and 81 photos captured by the motorized stepper system at a 5× magnification with 65 mm extension tubes, respectively. (**E**,**F**) Dorsal view images of an ant stacked from 91 images captured by the FACE system with a 4× objective lens at a 2.5× magnification and 51 images captured by the motorized stepper system at a 5× magnification with 65 mm extension tubes, respectively. (**G**,**H**) Lateral view images of the hoverfly stacked from 155 photos captured by the FACE system with a 4× objective lens at a 2× magnification and 81 photos captured by the motorized stepper system at a 4× magnification with 65 mm extension tubes, respectively. (**I**,**J**) Frontal view images of a honeybee stacked from 249 images captured by the FACE system at a 2.5× magnification and 166 images captured by the motorized stepper at a 2.5× magnification, respectively. Zoom-in inset images of the same image regions in every comparison are provided below each image.

**Table 1.** A comparison of the instrument cost between the current FACE and motorized stepper systems.

|  | FACE Method | Motorized Stepper Method |
| --- | --- | --- |
| Camera | 500 USD (4K CCD) | 500 USD (Canon 800D) |
| Lens | 100 USD (0.6–6× zoom lens); 100 USD (4× and 10× objective lens); 450 USD (DCR-150 + DCR-250 + MSN-202 + MSN-505 macro lens) | 500 USD (Laowa 25 mm ultra macro lens) |
| Extension tubes | - | 54 USD |
| Motorized rail | - | 299 USD (WeMacro) |
| Microscopic holder | 100 USD | 149 USD (WeMacro) |
| Helicon Remote control software (Lifetime license) | - | 48 USD |
| Helicon Focus stacking software (Lifetime license) | 100 USD | 100 USD |
| DIY 3D printed diffusion tunnel with LED light strip | 100 USD | 100 USD |
| Total price | 800 *–1450 ** USD | 1750 USD |

* The estimated cost of standard FACE setup not including extra macro lens. ** The estimated cost of upgraded FACE setup including extra macro lens.

## 4. Discussion

### 4.1. Advantages of the Current FACE System

#### 4.1.1. Shorter Operation Time and Capability to Capture Images of Living Objects

Since it is a video-based image capture method, the current FACE system has a significantly shorter operation time for acquiring photos for stacking, unlike most previous conventional methods. With the help of a stable frame capturing capability of 30 fps by the 4K CCD, 150–300 images on different focal planes can be obtained easily within 10 s, which is about 60- to 120-fold faster than the current motorized stepper method. Thus, by using the current FACE method, it is possible to do the image stacking of living animals if the subject maintains a non-moving posture for a few seconds. As an example, a stacked image based on a second-level video recording of a living lynx spider (*Oxyopes salticus*) was provided in this report (Video S1). These features also have overcome some limitations faced by other previously published low-cost setups, such as the limited number of images in a stack and lower versatility when it comes to specimen dimensions encountered by the prior setup demonstrated by Mertens et al. with compact cameras with focus stacking functionality [19]. Furthermore, since the current method is a video-based method, it enables users to obtain more images in a shorter time compared to the semi-automatic low-budget approach provided by Brecko et al. [20]. In addition, although other methods can also do an image stacking of living subjects, they require high-end cameras and lenses with built-in focus bracketing functions (e.g., Olympus EM1 Mark 2). This issue makes the current FACE method advantageous in terms of instrument setup cost. These features help users document large amounts of subjects and some aspects of specimens that are required to be processed alive, such as their behaviors.

#### 4.1.2. High Flexibility in Obtaining the Images of Objects from cm to mm Scale

While typical macro photography uses a 100 mm macro lens that can reach 1:1 or 2:1 magnification, some unique macro lenses with zoom functions, like Canon MP-E 65 mm and Laowa 25 mm, can offer extreme macro magnification up to 5×. Inspired by this design, a monocular lens with 0.6–6× magnification was used as a primary lens in the present study. Later, a macro lens or a long-working distance objective lens was mounted on it via thread adapters to elevate the final magnification. This combination allows users to conduct macro photography at a wide FOV range, ranging from 0.6 to 20 mm scale. In this range, it can cover the size of most minute animals such as ants (2 mm) and flies (5 mm), and moderately sized insects like bees (15 mm), beetles (7–15 mm, Figure S5), among others, highlighting the flexibility of the current FACE method than any conventional method, including like

the current motorized stepper method in terms of macro photography. However, one must remember that besides macro photography, a DSLR camera might have a much wider set of applications.

### 4.2. Limitations of the Current FACE System

#### 4.2.1. Unable to Control Aperture and Exposure Time

Since the current FACE method is based on a low-cost monocular zoom lens and CCD system, aperture and ISO settings are unavailable, unlike the current motorized stepper system. As users cannot control the opening of a lens's diaphragm, focus depth adjustment becomes impossible in the current method. In addition, camera-based conventional macro photography can also set the ISO setting and, thus, reduce image noise to obtain a better image resolution when the ISO value is set low. Meanwhile, the current CCD-based FACE method cannot manually adjust the ISO value. Therefore, although the current 4K CCD has an auto adjustment of exposure feature, this system will still show more image noise than the current motorized stepper method, especially at high magnifications.

#### 4.2.2. Relatively Short Working Distance at High Magnification

Although the current FACE system can achieve relatively high magnifications, some lenses reduce their working distance. For example, when a lens higher than $4\times$ magnification is mounted to the system, the working distance significantly decreases to only 10 mm. This relatively short working distance increases operational difficulties and substantially reduces the image quality due to insufficient light illumination. However, using long working distance objective lenses (e.g., Mitutoyo Infinity Corrected Long Working Distance) can overcome this limitation but with a relatively higher cost.

#### 4.2.3. Requires Additional Steps to Convert the Video into Images

Since the current FACE method is a video-based method, the output files are video files, and thus, before stacking the images, an additional step is required to convert the video file into an individual image file. Some software (e.g., Helicon Focus) provided this feature with an additional fee. Furthermore, a computer with relatively adequate specifications is recommended to reduce the operation time of this video conversion process.

#### 4.2.4. Heavily Relies on the User's Manual Observation Prior to a Video Recording

In the current motorized stepper method, Helicon Remote software was used to adjust the necessary parameters before starting the image capture process. This software has several features that can help users obtain the desired images, including over-exposure and focus indicators, which, unfortunately, are not available in the current FACE method since this method relies on the user's manual observation. In addition, since this method is conducted manually, the chance of obtaining inconsistent results is higher than with the motorized method.

### 4.3. Future Applications of the Current FACE System

Since the current FACE system had comparable results with the current motorized stepper method, we believe that this system also can be applied in various applications the current motorized stepper method is applied to and other works that require the photographic documentation of objects in small sizes, ranging from centimeters to millimeters. For example, this high-speed image stacking operation can help researchers in image documentation and digitalization of objects, especially small objects such as insects, in museums or research laboratories. In addition, it might also be potentially useful in 3D modeling by providing the images of the object layer by layer that could be processed later for modeling the 3D structure of an object, especially small objects, as the first step of a reconstruction of structure process [21–23]. Finally, besides being potentially used for various practical purposes in numerous fields of science, it can also be useful in the conservation of art, history, and natural science specimens.

## 5. Conclusions

In this paper, we reported a video-based method to conduct macro photography in a fast and cost-effective manner for the first time as an alternative method to do image stacking (standard operation protocol was provided in Video S2). The most promising advantage of the current FACE method is the relatively less time-consuming operating procedure with a comparable image results quality to the current image-based motorized stepper method. However, one also has to keep in mind that the current method also possesses several disadvantages need to be considered before deciding. In addition, continuous and uniform lights softened by a double-layer diffusion tunnel make macro photography feasible to be conducted without the aid of a flashlight. This high-speed image-stacking operation can be useful for scientists who routinely digitize images in their working places, such as in museums or research laboratories. In the end, the authors leave the readers to decide which method is more suitable for their work since both methods have advantages and disadvantages.

**Supplementary Materials:** https://www.mdpi.com/article/10.3390/inventions7040110/s1; Figure S1. They used macro and microscopic objective lenses for macro photography in this study. Figure S2. A 3D-printed diffusion tunnel was used in this study. Figure S3. Materials used to position objects. Insect pins of 0 or 00 sizes were used to adjust the object position. Figure S4. Materials used to conduct background replacement. Figure S5. Image stacks for *Adoretus formosanus* generated by using FACE setup. Video S1. Live stack for lynx spider by using FACE setup. Video S2. Standard operation protocol for FACE method.

**Author Contributions:** Conceptualization, G.A. and C.-D.H.; methodology, T.-W.H.; validation, M.-D.L.; formal analysis, G.A.; investigation, M.-D.L., T.-R.G. and C.-D.H.; resources, K.H.-C.C. and J.-C.H.; writing—original draft preparation, G.A. and C.-D.H.; visualization, C.-D.H.; supervision and funding acquisition, T.-R.G. and C.-D.H. All authors have read and agreed to the published version of the manuscript.

**Funding:** This research was funded by the Ministry of Science Technology, Taiwan, grant numbers 108-2313-B-033-001-MY3 to C.-D.H.

**Informed Consent Statement:** Not applicable.

**Data Availability Statement:** The data presented in this study are available on request from the corresponding author.

**Acknowledgments:** We thank Jeroen van Gastel and Giuseppe Criseo for providing professional comments on macro photography. We also thank Suhaib Firdous Yatoo and Ethan J. Beckler for offering very useful advice and suggestions for developing the FACE technique and Roi Martin B Pajimna for critical reading and paper editing.

**Conflicts of Interest:** The authors declare no conflict of interest. The funders had no role in the design of the study, in the collection, analyses, or interpretation of data, in the writing of the manuscript, or in the decision to publish the results.

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
