# Peer review of "A Fast and Cost-Effective (FACE) Instrument Setting to Construct Focus-Extended Images"

_inventions, doi:10.3390/inventions7040110_

Round 1
Reviewer 1 Report
Image stacking is a crucial method for micro or macro photography. It captures images at different focal planes and then merges them into a single, all-in-focus image with extended focus. This method has been extensively used for digital documentation by scientists working at museums or research institutions.
1. The format of some cited references is not standard and unified and some of the references are conference paper.
2. The linguistic quality needs improvement. There are some grammatical errors, which significantly affects the quality of the paper.
3. Some other algorithm can be used as test algorithm, the inspiration of your work must be highlighted.
3. Some of the figures are not clear enough.
5. More experiments should be given to verify the proposed model.
Author Response
Comments and Suggestions for Authors
Image stacking is a crucial method for micro or macro photography. It captures images at different focal planes and then merges them into a single, all-in-focus image with extended focus. This method has been extensively used for digital documentation by scientists working at museums or research institutions.
- The format of some cited references is not standard and unified and some of the references are conference paper.
Thank you for the reminder. The authors checked the format of the references and verified it by using EndNote software with the “MDPI” set as the style. The authors hoped that the updated reference format is unified in this updated manuscript. In addition, regarding the conference paper, to the best of the authors’ knowledge, the journal allowed this type of reference as mentioned in the Inventions’ website, specifically in the instructions section. Therefore, the authors believed that this type of reference could be cited in the manuscript. We appreciated your professional comments.
- The linguistic quality needs improvement. There are some grammatical errors, which significantly affects the quality of the paper.
The authors appreciated the suggestion and tried their best to correct the grammatical errors in the previous version of the manuscript. The authors hoped that the improvement could enhance the quality of this revised paper. We appreciated your professional comments.
- Some other algorithm can be used as test algorithm, the inspiration of your work must be highlighted.
Thank you for the suggestion. However, the authors do not fully acknowledge what kind of algorithm that can be used as a test. Therefore, the authors appreciate it if the reviewer would give some explanation regarding the mentioned algorithm so the authors could understand the direction given by the reviewer.
4. Some of the figures are not clear enough.
The authors thanked the reviewer for the comment. However, the authors used the highest-quality figures in the manuscript. If the reviewer referred to the images in high magnification, the lack of clarity in the images might come from the original resolution itself, which could not be enhanced further and needed to be maintained in order to highlight the output quality of each method. Nevertheless, if this is not what the reviewer refers to, the authors hoped that the reviewer could point out which figures that are not clear enough so the authors could do a revision correctly.
- More experiments should be given to verify the proposed model.
In previous submitted version, the supplymentary data file might be missing due to format conversion step. In this revised paper, we provided the detail supplymentary data again and added one more extra data in the supplymentary data Figure S5 to show the different views of bettle Adoretus formosanus generated by FACE setup.
Reviewer 2 Report
This paper proposes the a Fast and Cost-Effective, called (FACE), instrument to construct focus-extended images. The paper is very well written and presented. I have some questions
1) please better describe the applications that this instrument can be applied in some paragraphs
2) please show that this instrumentation is better than current structured (just put a discussion)
3) can this method be applied for 3D modelling and potential 4d (3d geometry plus time). For instance for cultural heritage applications
Rodríguez-Gonzálvez, Pablo, et al. "4D reconstruction and visualization of cultural heritage: Analyzing our legacy through time." The International Archives of Photogrammetry, Remote Sensing and Spatial Information Sciences 42 (2017): 609.
Johnson, P. S., et al. "Online 4D reconstruction using multi-images available under Open Access." ISPRS Annals of the Photogrammetry, Remote Sensing and Spatial Information Sciences (2013).
Kacprzyk, Zbigniew, and Tomasz KÄ™pa. "Building information Modelling–4D Modelling technology on the example of the reconstruction stairwell." Procedia engineering 91 (2014): 226-231.
Author Response
Comments and Suggestions for Authors
This paper proposes the a Fast and Cost-Effective, called (FACE), instrument to construct focus-extended images. The paper is very well written and presented. I have some questions
1) please better describe the applications that this instrument can be applied in some paragraphs
The authors appreciated the comment. It is true that the application of the current setup was not described in detail in the previous version of the manuscript. Therefore, in the updated version, the addition section in the Discussion part was added to mention the applications of this method, which includes image documentation and digitalization of objects that can be useful for 3D modeling, conservation of art, history, and natural science specimens, and various practical purposes in numerous fields of science.
2) please show that this instrumentation is better than current structured (just put a discussion)
Thank you for the suggestion. However, based on the present test’s results, the authors cannot definitely claim that the current setup is better than the commercial setup. Nevertheless, the current setup has indeed, several advantages over the commercial setup with also some limitations. As mentioned in the manuscript, specifically in the discussion section, the current setup provides a shorter operation time and capability to capture images of living objects and high flexibility in obtaining the images of objects from cm to mm scale. However, by using this setup, users would be unable to control aperture and exposure time and are required to work in a relatively short working distance at high magnification with results that heavily relies on the user’s manual observation prior to a video recording. Based on these results, the authors leave the decisions to readers, which depends on what it will be applied to since each method has its own advantages and disadvantages. In addition, the advantages of the current instrument over the current commercial method, it is already described in detail in the discussion part (section 4).
3) can this method be applied for 3D modelling and potential 4d (3d geometry plus time). For instance for cultural heritage applications
Rodríguez-Gonzálvez, Pablo, et al. "4D reconstruction and visualization of cultural heritage: Analyzing our legacy through time." The International Archives of Photogrammetry, Remote Sensing and Spatial Information Sciences 42 (2017): 609.
Johnson, P. S., et al. "Online 4D reconstruction using multi-images available under Open Access." ISPRS Annals of the Photogrammetry, Remote Sensing and Spatial Information Sciences (2013).
Kacprzyk, Zbigniew, and Tomasz KÄ™pa. "Building information Modelling–4D Modelling technology on the example of the reconstruction stairwell." Procedia engineering 91 (2014): 226-231.
The authors appreciated the reviewer for the question and for providing some useful literature that could enhance the quality of the current manuscript. Since the authors’ background is not closely related to structure modeling, the authors tried their best to comprehend this field and the current method’s potential application in this field. After reading the provided journals, the authors believed that the current system could be applied to the modeling process, especially 3D modeling since generally, the reconstruction of structures process is started with provided documentation of the object. Thus, the current setup could be an alternative method in this step by providing the images of the object layer by layer that could be processed later for modeling the 3D structure of an object, especially small objects. This potential application of the current method was added to the manuscript. In addition, the provided references were also cited in the paper.
Round 2
Reviewer 2 Report
All m comments are OK.